# Significance of the Diagnosis of Executive Functions in Patients with Relapsing-Remitting Multiple Sclerosis

**DOI:** 10.3390/ijerph181910527

**Published:** 2021-10-07

**Authors:** Aneta R. Borkowska, Beata Daniluk, Katarzyna Adamczyk

**Affiliations:** 1Department of Clinical Psychology and Neuropsychology, Institute of Psychology, University of Maria Curie-Sklodowska, 20-612 Lublin, Poland; beata.daniluk@mail.umcs.pl; 2Green Clinic, Center for Treatment and Rehabilitation of Psychosomatic Disorders and Addictions, 20-030 Motycz, Poland; andre172@wp.pl

**Keywords:** cognition, executive functions, multiple sclerosis, depression, neuropsychological assessment

## Abstract

Multiple sclerosis (MS) is a progressive chronic disease of the Central Nervous System (CNS). Cognitive decline occurs rather rarely in relapsing–remitting multiple sclerosis (RRMS) compared to other types. The present study aimed to assess executive functions (EF) in relation to clinical and demographic variables in patients with RRMS. The study involved 22 individuals with RRMS (aged 23 to 49 years) and 22 matching controls. All the individuals with RRMS were in the remission phase. The assessments were carried out using MoCA, BDI-II, Halstead Category Test, Porteus Maze Test, verbal fluency tasks and Stroop Colour-Word Interference Test. The findings show that the two groups differed significantly in all the tests. All patients with RRMS in the remission phase presented at least one cognitive deficit, observed in general cognitive functioning, abstract reasoning or other executive functions, i.e., fluency, interference suppression, planning, or ability to modify activity in response to feedback. The deficits in most cases (except for those measured with the MoCA, Category Tests and phonemic fluency), are not related to intensity of depression and duration of the disease. Findings suggest that the diagnostic process in the case of patients with RRMS may include psychological assessment focusing on potentially existing cognitive, mainly executive, deficits and their severity.

## 1. Introduction

Multiple sclerosis (MS) is a progressive demyelinating and neurodegenerative disease of the CNS, and a constitutive cause of disability in patients of varied ages; it most commonly starts in early adulthood (20–40 years), and its aetiology is still unknown [1,2]. This is a chronic disease, and its clinical neurological picture is linked to the location of the pathological changes in CNS (most commonly occurring in multiple areas) and to the variant of the disease: relapsing–remitting (RRMS), primary or secondary progressive (PPMS or SPMS) as well as the phase of the disease (relapse vs. remission) [2]. The symptoms vary in terms of severity and dynamics of the progress. In addition to neurological symptoms, many patients present difficulties in cognitive functioning, which further affects the quality of their lives [3]. The percentage of patients with signs of cognitive impairment reported in different studies is varied, ranging between 25% and 70%, depending on the specific characteristics of the condition, as well as approach to recruitment (hospitals, rehabilitation centres versus population studies) and adopted definition of cognitive impairments [1,3,4]. It has been suggested that the progressive variant of MS leads to more severe cognitive impairments than RRMS [5]. Ruano et al. reported that, relative to the disease variant, these types of impairments were observed in 34.5% of patients with CIS (clinically isolated syndrome), 44.5% of patients with RRMS, 79.4% of patients with SPMS and 91.3% of patients with PPMS. The information processing speed was the most commonly affected domain in all the groups. Significant differences were observed between relapsing and progressive forms of the disease but not between CIS and RRMS or SPMS and PPMS patients [6]. A meta-analysis of 47 studies, which reported findings related to a total of 4460 patients, established that patients with PPMS present slightly more severe impairments in all the cognitive domains than patients with RRMS; deficits are more prominent in verbal learning, processing speed and verbal memory than in other domains [7]. Cognitive deficits appear rather early; they are persistent and more commonly progressive in nature. It has been pointed out that cognitive changes may potentially be used as a marker of the disease progress [2]. The currently conducted neuropsychological studies, in combination with fMRI findings, allow us to better understand cognitive problems experienced by patients with MS, both young individuals at the initial stage of the disease and those at later stages [2]. Severity of cortical atrophy, i.e., the size of the lesion, correlates with the gravity of cognitive changes; the size of the third ventricle is significantly related to the cognitive level, and the gravity of cognitive impairments is to a greater degree linked to subcortical atrophy, including thalamic lesions, than to the size of atrophy in the whole brain [2,8,9]. The glutamate concentration in the gray matter is linked to memory impairments [10]. 

MS adversely affects many aspects of patients’ cognitive functioning, including attention, effectiveness of information processing, executive functions, processing speed, long-term memory, visual learning and recall [2,3,4,5]. The problems diagnosed most frequently, at a rate of 51.9–54.3%, are related to processing speed, visual learning and memory. Cognitive function domains less commonly affected by MS include simple attention processes (e.g., digit span tasks) and basic language processes, such as naming and comprehension [5,11]. A comprehensive study focusing on function impairments in patients with MS, carried out with the use of the Minimal Assessment of Cognitive Function in MS, showed that 28–52% of patients present decreased processing speed, and 30–55% present memory impairments assessed with CVLT-II and BVMT-R [12]. Explicit memory, assessed using tasks involving learning and recall, as well as episodic memory of facts and events are also frequently impaired. Semantic memory (related to words and symbols) as well as implicit memory (involving no conscious focus on remembering) are usually maintained. Other cognitive problems are related to executive functions, verbal fluency, and visuospatial analysis [4]. Patients also report difficulties if they are required to simultaneously perform multiple tasks and search for words. The evidence related to overall intelligence level is inconclusive. Some studies show that IQ seems to be unaffected by MS, while others suggest there is a decline, sometimes subtle but significant. Dementia occurs rarely in patients with MS; mild cognitive deficits are far more frequently diagnosed [5]. 

Executive functions involve cognitive processes necessary in intentional behaviours and in effective adaptation to changes and requirements occurring in the environment. They include abilities to make plans, to anticipate effects of actions and to manage resources. Deficits are reflected by problems in abstract and conceptual thinking, by poor fluency, decreased planning skills and poor organisation of activity and operational memory. Deficits in this sphere are significantly less common in patients with MS than processing and memory deficits. Drew et al. found that 17% of patients have problems in the broadly defined sphere of executive functions (shifting, inhibition and fluency) [13]. Henry and Beatty observed that patients with MS have considerable problems performing both semantic and phonemic fluency tests [14]. According to these researchers, the level of difficulty in fluency tasks is a sensitive measure of neurological impairments in MS. Likewise, a high number of perseverative errors is typical for MS. Working memory is usually poorer, just like such complex aspects of attention as selectivity, divisibility and shifting [4]. Measures of executive function (EF) were found to be correlated to depression level in patients with MS [15].

The present study mainly aimed to assess executive functions in patients with RRMS as compared to matched healthy controls. The analyses were also designed to identify the relationships between the level of executive functioning in patients with MS and clinical as well as demographic variables. 

In the study, it was hypothesised that individuals with RRMS present poorer executive functions than healthy individuals. It was anticipated that executive functions are correlated to the general level of cognitive functioning and to severity of depression symptoms.

## 2. Materials and Methods

The study involved 44 individuals (40 females and 4 males) aged 23–49 years (*M* = 37.66; *SD* = 6.99), recruited into clinical and control groups. The first one consisted of individuals with relapsing–remitting multiple sclerosis—20 females and 2 males (mean age *M* = 37.68; *SD* = 7.05). Duration of the disease ranged from one to 21 years (*M* = 9.64; *SD* = 5.91). The majority of the patients were residents of urban areas (72.7%) and were in relationships (86.4%). In the study group, there were 10 individuals with secondary and 12 with higher education. All the individuals with MS were in the remission phase. They volunteered to participate in the study in response to an announcement posted on websites dedicated to individuals with multiple sclerosis. The control group consisted of subjects matched for age, sex and education to the individuals with MS. Hence, there were 20 females and 2 males (mean age *M* = 37.64; *SD* = 7.10). The majority of the subjects were residents of urban areas (81.8%) and were in relationships (72.7%). One in two control subjects reported secondary education (11 individuals), the rest reported higher education. No differences were found between the patients with MS and the controls as regards the place of residence (χ^2^ = 0.518; *p* > 0.05) and education (χ^2^ = 1.257; *p* > 0.05). Three individuals with head injuries, physical health problems and addiction to psychoactive drugs, possibly impairing executive functions, were excluded from the study. 

Executive functions were assessed using the Halstead Category Test, part of the Halstead-Reitan Neuropsychological Test Battery, Porteus Maze Test (PMT), verbal (phonological and semantic) fluency tasks and the Stroop Colour-Word Interference Test (SCWT).

The Halstead Category Test (HCT) is recognised as a sensitive neuropsychological measure of brain dysfunctions. It enables assessment of various cognitive functions, including the abilities to formulate abstract concepts (abstract reasoning) and to apply feedback to modify one’s behaviours, as well as learning and memory capacities and the ability to maintain attention while performing a lengthy task [16]. It is a measure of abstract attitude, as an ability of conceptual reasoning and planning, irrespective of the type and sensory modality of the material [17]. The overall performance indicator in the test was the number of errors.

The Stroop Colour-Word Interference Test (SCWT) enables assessment of cognitive inhibition in tasks involving interference caused by an automatic activity performed simultaneously; hence, it measures inhibitory control in a conflict situation. Accurate performance in the task depends on the ability of cognitive inhibition and on resistance to interference, in a situation when processing of one feature of a specific stimulus affects simultaneous processing of another feature of the same stimulus [18]. A computerised version of the test, used in the study, comprised a series of 10 trials. In each trial, squares in three colours (red, green and blue) can be seen at the bottom of the screen. They indicate the right way to respond. At the start of each trial, in the middle of the screen, the subject can see a word (name of colour) printed in one of the three colours of ink. The subject is asked to respond, by pressing the key corresponding to the actual colour of the print rather than to the name of the colour. They are to press the keys “left arrow”, “down arrow” or “right arrow” corresponding to the colour red, green or blue. For instance, if the word “red” appears printed in the colour blue, in accordance with the clue at the bottom of the screen, they should press the right arrow key. The responses should be given as quickly as possible and accurately. The number of errors and time needed to complete the task were used as performance indicators in this test [19].

The Controlled Oral Word Association Test (COWAT), which is a verbal fluency test, assesses accessibility of words in spontaneous speech production; it is a good measure of mental productivity and language processes. The tasks require adequate strategies as well as effective retrieval from long-term memory, inhibitory control and working memory. The study used two verbal fluency tasks. In the first one, i.e., a phonological fluency task, the subjects were asked to produce as many words as possible starting with the letter K. They could not use proper names, or words with the same root and different endings (e.g., *król, królowa, królewna*). In the other one, the category fluency task, the subjects were to produce as many words as possible in the category “animals”. Each task was continued for a duration of 60 seconds. The performance indicators in the task correspond to the number of words produced and meeting the criterion, and number of errors which comprise repetitions and words failing to meet the criterion.

The Porteus Maze Test (PMT), which is a paper–pencil device, consists of a series of mazes, arranged according to increasing level of difficulty. It mainly enables assessment of the abilities of anticipation and planning. The subject is to exit the maze by drawing the road which is to be travelled; in this process, they must avoid entering dead-ends and crossing the line of the maze. The study used 10 mazes of increasing difficulty, printed on separate pieces of paper. The subjects were to correctly solve all the mazes in the shortest possible time. The assessment took into account the time needed to perform the tasks and the number of errors.

The Montreal Cognitive Assessment (MoCA) test is a screening tool used for the needs of early diagnosis of mild cognitive impairments of various aetiology, manifesting in such areas as short-term memory, visuo-spatial abilities, executive functions, attention, concentration and working memory, language skills and spatial and temporal orientation. The total score in the whole test was used as the indicator of general cognitive capacity (max. 30). A score of 24 points or more is thought to correspond to normal status in this domain [20].

In addition to cognitive functions, the present study investigated the intensity of depression symptoms; this measurement was performed using the Beck Depression Inventory—Second Edition (BDI-II). The method is frequently applied in research focusing on multiple sclerosis, as it enables self-reported assessment of depression symptoms. BDI-II is applied as a measure of depression symptom intensity; however, it cannot be treated as an assessment tool providing a basis for diagnosis of clinical depression [21].

### Statistical Analysis Methods

The analyses were computed using the software package SPSS IBM Statistics 26, PS IMAGO 6 (Predictive Solutions, Kraków, Poland, license number 1142). At the first stage, the group of individuals with MS and the controls were compared with regard to the executive functions and general cognitive processes; the related analyses were performed using the Student’s t-test for two independent samples in the case of variables with normal distribution, and the nonparametric Mann-Whitney U-test in the case of variables whose distribution differed from normal distribution. Effect size was determined using Cohen’s d as well as the Glass rank-biserial correlation coefficient (r_g_). At the second stage, profile analysis of all the individuals with MS was carried out, taking into account their performance in the specific subtests in order to determine the prevalence of executive dysfunctions in this group. For this purpose, raw scores were transformed into standardised results, based on the means and standard deviations of the results acquired by the healthy controls. Subsequently, performance in each test was assessed, taking into account the criterion of standard deviation: for the indicators expressed by the number of points (correct responses)—correct performance (−1.5 ≤ *z* ≤ 1.5), impaired performance (*z* < −1.5) and for the indicators expressed as the number of errors and performance time—correct performance (−1.5 ≤ *z* ≤ 1.5), impaired performance (*z* > 1.5) [22] (cf., Treder, Jodzio, 2013). The relationships were examined using Spearman's rank correlation coefficient. The level of α < 0.05 was assumed in the calculations.

## 3. Results

The descriptive statistics of the results and comparison of the group of patients with RRMS and the controls in the specific test measures are shown in Table 1.

The individuals with MS presented significantly poorer performance in all the tasks assessing executive functions, compared to the controls. Frequency of the deficits is shown in Table 2.

Depending on the task and the process assessed (see Table 2), deficits were observed in the subjects at a rate ranging from 9.1% to 81.8%. The problems were most frequently observed in the semantic fluency task and in the Category Test. Further analyses focused on signs of cognitive decline, which, in line with the assumptions, would be reflected by a score in at least one measure that was 1.5 standard deviations lower than the mean score in the control group; the related findings show that performance reflecting executive dysfunction defined in this way was presented by all the patients. Analysis of the distribution of the number of executive dysfunctions in the individuals with RRMS showed that as many as 22.7% of the subjects presented a decline in four measures, whereas one in two subjects presented problems in 5 or more aspects (on average the patients had deficits in five measures). Two patients were found with lower scores in eight measures (i.e., with the high rate of 89%). 

Subsequent analyses focused on the MoCA rates, depression symptoms and duration of the disease in years in relation to measures of executive functions. The results are presented in Table 3. Since significant correlations were identified only for three measures, shown in the Table 3, the other measures were disregarded.

The findings show that longer duration of the disease correspond to lower scores in MoCA and higher rate of errors in the Category Test, as well as poorer performance in phonemic fluency tasks, reflecting more serious cognitive deficits. Intensity of depression symptoms negatively correlates with the scores in the MoCA and positively with the Halstead Category Test, whereas MoCA negatively correlates with the Halstead Category Test. Additionally, the findings show significant, high correlation between BDI-II and duration of the disease (R = 0.725, *p* < 0.001).

## 4. Discussion

In the recent years, researchers have reported a growing body of evidence related to cognitive impairments occurring in MS; however, it has been suggested that individuals with RRMS, in comparison to those with other types of MS, less frequently present cognitive problems [6]. The present findings confirm that MS is associated with cognitive deficits, which is consistent with earlier reports. Studies by Ochi et al. [23] and Shulz et al. [24] showed that, on average, 70% of patients with SM present cognitive deficits, particularly at the early stages of the disease. Lower rates were reported by Trenova et al., who found cognitive deficits in 25–40% of patients with MS [25]. A large study, involving 1040 patients with MS, reported deficits occurring at a rate of 46.3% [6], whereas individuals with RRMS, the type investigated in the current study, presented such problems at a rate ranging between 45% and 51% [6,26]. The present findings do not confirm the reports, suggesting that cognitive problems are less common in patients with RRMS than in patients with other forms of MS. To the contrary, all the subjects in our clinical group presented at least one cognitive deficit, in line with our definition reflected by a score that was 1.5 standard deviations lower than the mean score in the control group. Notably, all the subjects were in the remission phase, yet their cognitive status differed significantly from that presented by the controls. The varied results reported in the literature may be associated with the fact that patients with various types of MS were assessed, whereas the differences in the rates of deficits in patients with RRMS may be linked to the fact that different neuropsychological tests were employed [26]. If MoCA or MMSE are the primary tools applied to assess cognitive functions, there will be few patients with low scores because of the screening nature of these tests, and because they are fairly easy [4,27,28]. Due to this, they do not capture subtle cognitive changes. MoCA provides information about patients’ overall cognitive functioning, and according to some authors, it presents sufficient sensitivity and specificity to enable diagnosis of deficits in MS [27,28]. However, other researchers argue that MoCA should not be recommended because the tool assesses cognitive domains which are different than those mainly affected in patients with MS [4,29]. The tools applied in the present study included MoCA as well as other tests evaluating executive functions; in fact, the latter tests identified more problems affecting the patients. The scores in MoCA show significant differences in the performance between the clinical group and the controls, and these are suggestive of general cognitive decline in RRMS, which is consistent with results reported by other authors [27,28]. However, the MoCA test identified deficits in the patients at a rate of only 36.4%, while other cognitive problems were found in all the patients. Hence, by applying more sensitive tools, including tests assessing executive functions, like in our study, it is possible to capture a larger number of cognitive deficits in RRMS. Another important reason for the differences in the measures of cognitive deficits may be linked to the different definition of cognitive impairments and criteria applied in distinguishing these. As a rule, a patient is classified in the group with impairments when at least in one [30] or in two tests applied [26], his/her scores are 1.5 standard deviations lower than the mean in the control group. As mentioned earlier, we applied the former approach [30]. 

Cognitive impairments in MS are varied. The most affected spheres include various aspects of attention, processing speed, memory, executive functions and visuospatial functions [24,26,31]. The present study focused on such executive functions as planning, interference suppression, abstract reasoning, ability to modify activity in response to feedback and verbal fluency. 

The results of all the tests applied in the study show that the performance of the individuals with RRMS in all the tasks assessing executive functions was significantly poorer in comparison to the control group. The largest group of patients (81.8%) presented deficits in the semantic fluency task, as reflected by the number of words produced. In the other indicators of fluency (number of produced words starting with the letter K, and number of repetitions), deficits were shown by 65% of the patients. Fluency tasks are considered effective probes for executive function, and are commonly included in neuropsychological batteries that assess such executive skills [32]. Hence, the present findings confirm that this is a sensitive measure of cognitive problems in RRMS. Verbal fluency is defined as the ability to rapidly and effectively produce words matching the specified phonological or semantic criterion. The deficits identified in this domain may adversely impact effective communication, consequently leading to limitations in the patients’ social life [33]. Tests of verbal fluency require search, access, selection, retrieval and pronunciation of as many words as possible in a restricted time period, based on a predefined criterion. This process may fail due to deficits in any of these cognitive components [34]. The high rate of such deficits in patients with RRMS in the present study is consistent with earlier research reports. This is particularly confirmed by a meta-analysis of 35 publications, taking into account a total of 3673 patients with MS, where it was demonstrated that study participants showed considerable problems in phonemic and semantic fluency tests. These deficits were more pronounced than the deficits in the measures of verbal intelligence and other commonly used tests assessing executive functions (e.g., WCST). Verbal fluency tests may be recognised among the most sensitive neuropsychological measures of cognitive impairments in MS [14]. Verbal fluency deficits have also been confirmed by neuroimaging studies. Significant correlations were shown between fluency measures and mean fractional anisotropy (FA) in several pathways [33]. 

The Stroop Test (SCWT) also differentiated the two groups, patients with MS and healthy controls, reflecting the fact that the former experience problems in interference suppression, an important executive function. This is consistent with results of other studies [35] which reported that patients with MS achieved significantly poorer scores in this test. However, Macniven et al. in their article discuss the mechanism, possibly explaining the poorer scores, and they suggest that it is not the actual deficits in executive functions but rather the more basic slowing down of the reaction that poses the main problem. Indeed, while comparing the scores in SCWT and in other tests assessing executive functions, the authors identified a common mechanism of difficulty in the measures applied, namely that of longer reaction time, which shows that this is definitely an important element of the mechanisms of difficulty. Similarly, in our study time needed to complete the task was significantly longer in the clinical group. However, Macniven et al. did not take into account the number of errors but only the reaction time indicator. The current findings show that the number of errors also differentiates the two groups, which suggests it is possible that two mechanisms are involved: slowing down and executive deficits. The percentage of individuals in the clinical group showing poorer performance in the Stroop Test as regards the number of errors (59.1%) was the same as the percentage of individuals presenting deficits in reaction time. A study which applied the Stroop Test in a dual task paradigm (motor and cognitive tasks–Stroop) showed that individuals with MS present higher costs when performing two tasks simultaneously, which is suggestive of executive deficits (divisibility of resources) [36]. Notably, assessment performed with the Stroop Test may clearly be of importance in practice; researchers have shown that remission of benign multiple sclerosis (B-MS) may be predicted with 82% likelihood, based on the score in the Stroop Tests [37]. Hence, the tool may provide support in reaching a differential diagnosis. 

The planning function was assessed in the current study using the Maze Test. The findings show that the two groups are differentiated by the number of errors but not by the time needed to complete the task. This reflects deficits in planning ability presented by patients with RRMS. This is consistent with results of other studies focusing on planning abilities, in which the Tower of London (TOL) test was used [38]. However, the present findings, contrary to those reported by Arnett et al., do not show decreased speed in the task performance, which suggests that it is mainly executive dysfunctions rather than only the slowing down that are characteristic for the patients with RRMS. 

Abstract reasoning, also investigated in this study, was assessed using the Halstead Category Test. Like in the other functions, in this case, the subjects with RRMS presented significantly poorer performance than the controls. Deficits were observed in 72.7% of the individuals in the clinical group. This means that problems related to abstract reasoning occur in patients with RRMS, which is not consistent with other research reports. Available evidence suggests that decline in this domain is observed mainly in progressive types of MS [7]. Our findings, however, show that patients with RRMS also present decline in abstract reasoning. The fact that such significant number of the patients experienced difficulty while performing the Category Test may be associated with the complex requirements of the test. The score in this test makes it possible to draw conclusions not only about abstract reasoning but also about the ability to use feedback, and about learning, memory and efficiency of attention processes. Patients with MS present deficits in all of these areas. 

In order to achieve more comprehensive understanding of executive deficits in RRMS, we also examined the relationship between MoCA and other executive function measures. The findings show that MoCA correlates with the Category Test results, and it does not correlate with other measures of cognitive functions. This suggests that individuals achieving high scores in MoCA, which reflect a lack cognitive impairments, may face problems performing tests assessing executive functions. Therefore, it seems that the MoCA should not be used as the only tool in diagnosing cognitive difficulties in MS. 

Finally, the current study investigated the relationship between executive functions and intensity of depression symptoms as well as duration of the disease. Many studies point out that depression affects cognitive function, including executive functions [38,39]; however, other studies suggest there is no correlation between depression indicators and cognitive measures [3,40]. Notably, however, conclusions about the effect of depression on cognitive functions, such as attention and executive functions, cognitive flexibility, verbal fluency, set-shifting, suppression, working memory and planning are frequently drawn based on comparative assessments of patients with diagnosed unipolar depression and with no symptoms, rather than with isolated symptoms [41]. Studies reported that approximately 30% of patients with MS present symptoms of depression particularly affecting speed of information processing, executive functions, attention, memory and motor functions [15,42]. In the present study, depression symptoms of moderate and high intensity were identified in 72.7% and in 22.7% of the patients, respectively. The findings show that the level of depression symptoms correlates negatively only with general cognitive assessment (MoCA) in the patients with MS (R = −0.519) and with scores in the Category Test, measuring abstract thinking as well as memory processes and attention (R = −0.591). No correlations were identified between the other executive function measures and intensity of depression symptoms. Hence, it appears that some cognitive functions are related to symptoms of depression, whereas others are not. This may depend on the type of function investigated. Various theories have been proposed to link depression with executive functions. From the cognitive viewpoint, the problem is explained by the resource allocation model [43]. In accordance with this approach, depression utilises some of the limited cognitive resources due to which it is impossible to engage sufficient resources in tasks requiring special effort. Our patients did not have a clinical diagnosis of depression and severe symptoms; they only presented elevated scores in the depression scale. Hence, the effect of the depression level on task performance was visible only in the most complex cognitive tasks. 

## 5. Conclusions

In summary, the current findings show that all the patients with RRMS present cognitive deficits (at least one), which are observed in various domains, in general cognitive functioning assessed with MoCA and in abstract reasoning and other executive functions, such as fluency, interference suppression, planning, and ability to modify activity in response to feedback. In most cases, these are not associated with the level of depression or the duration of the disease. Only the general cognitive status (MoCA) and abstract reasoning (the Category Tests) were found to be related to depression symptoms and duration of the disease, whereas one executive function, i.e., phonemic fluency is linked to the duration of the disease. Obtained results suggest that, in patients with RRMS at early stages of the disease, the psychological, cognitive diagnosis would be recommended for possible cognitive, mainly executive, limitations. Since cognitive deficits adversely affect patients’ quality of life, if these are diagnosed — it is necessary to implement early therapies enabling recovery of these functions [2,44]. However, such recommendations require confirmation of the obtained results by other studies.

## Figures and Tables

**Table 1 ijerph-18-10527-t001:** Descriptive statistics of the results and comparison of the variables investigated in the group of patients with RRMS and the control group.

Test	Clinical Group (N = 22)		Control Group (N = 22)		Z/t	*p*	r_g_/ Cohen’s d
	M SD (Min–Max)		M SD (Min–Max)				
BDI-II	12.14 (0–32)	9.01	0.09 (0–1)	0.29	−5.72 (Z)	< 0.001	0.95 (r_g_)
MoCA	26.82 (21–30)	2.81	29.91 (29–30)	0.29	−4.66 (Z)	< 0.001	0.74 (r_g_)
Halstead Category Test	64.32 (18–133)	27.11	30.23 (10–60)	13.62	5.27 (t)	<0.001	1.59 (r_g_)
Phonemic fluency–number of words	17.23 (8–28)	5.81	24.91 (17–37)	4.29	−4.99 (t)	<0.001	1.50 (d)
Phonemic fluency–repetitions	1.91 (0–4)	0.92	0.77 (0–2)	0.69	−3.89 (Z)	< 0.001	0.65 (r_g_)
Semantic fluency–number of words	20.68 (15–37)	5.57	29.32 (25–33)	2.36	−4.77 (Z)	< 0.001	0.84 (r_g_)
Semantic fluency–repetitions	1.86 (0–4)	1.13	0.82 (0–2)	0.85	−3.07 (Z)	0.002	0.52 (r_g_)
Stroop Test–time	1054.0 (471–1577)	335.95	680.23 (402–1399)	219.70	−3.71 (Z)	<0.001	0.65 (r_g_)
Stroop Test–errors	1.91 (0–6)	1.54	0.18 (0–3)	0.66	−4.57 (Z)	<0.001	0.72 (r_g_)
Maze Test–time	232.27 (95–648)	139.09	236.18 (113–419)	78.16	−1.17 (Z)	0.240	–
Maze Test–errors	12.91 (4–26)	6.47	5.09 (0–12)	3.78	4.89 (t)	<0.001	1.48 (d)

M-mean; SD-standard deviation; *p*-significance level; z statistics based on Mann-Whitney U test; t statistics based on Student’s *t*-test; r_g_ Glass’ coefficient; d Cohen’s d coefficient.

**Table 2 ijerph-18-10527-t002:** Frequency of deficits identified in the specific trials.

Test	Indicator	Frequency of Deficits	%
MoCA	Score	8	36.4
Halstead Category Test	Number of errors	16	72.7
Stroop Test	Time	13	59.1
	Number of errors	13	59.1
Maze Test	Time	2	9.1
	Number of errors	12	54.5
Phonemic fluency	Number of words Number of errors	14 15	63.6 68.2
Semantic fluency	Number of words Number of errors	18 5	81.8 22.7

**Table 3 ijerph-18-10527-t003:** Correlations in the group of patients with MS between duration of the disease, MoCA, BDI-II and Halstead Category Test and verbal fluency.

Variable	MoCA		BDI-II		Duration of Disease	
	R	*p*	R	*p*	R	*p*
MoCA	–	–	−0.519 *	0.013	−0.510 *	0.015
Category Test	−0.586 **	0.004	0.591 **	0.004	0.564 **	0.006
Phonemic fluency- number of words	0.157	n.s.	−0.334	n.s.	−0.499 *	0.018

* significant result with *p* < 0.05; ** significant result with *p* < 0.01; n.s. not significant.

## Data Availability

The data presented in this study are available on request from the corresponding author. The data are not publicly available due to ethical issues. A consent provided by participants included assurances that their results would not be publicly available.

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
