# Peer review of "Significance of the Diagnosis of Executive Functions in Patients with Relapsing-Remitting Multiple Sclerosis"

_ijerph, 2021, doi:10.3390/ijerph181910527_

Round 1

Reviewer 1 Report

The investigators compare a sample of patients with RRMS with controls on a small battery of mainly executive dysfunction-sensitive tests with controls and discuss their findings. Table 1 I looks to have a transposition of semantic fluency words vs errors. I was also puzzled by the apparently greater discrimination of phoemic fluency, as suggested by the numerically higher Cohen’s D for it, relative to semantic fluency, yet we are also told that 82% of patients had abnormal scores on category fluency while only 63% had deficits on phonemic fluency. Is there an error here, or can the investigators otherwise explain this apparent anomaly? On the basis of the findings, the investigators recommend that ‘patients with RRMS should be assessed at early stages of the disease for possible cognitive, mainly executive limitations’. While it is certainly appropriate for MS patients’ executive function to be evaluated, it seems odd to justify this recommendation based on a study that, apart from the MoCA focused ONLY on executive function. In other words, this study did not demonstrate a particular salience of executive dysfunction because data on other functions was lacking. The fact that so many executive function tests were included is one factor that could inflate the rate of apparent executive dysfunction. The known difficulty/sensitivity of the Halstead Category Test, typically not included in studies of MS patients is another. Yet another possibility is that the sample itself was biased. To that end, did the way in which the study was advertised suggest the possibility of selective recruitment of those on the more severe end of the cognitive impairment spectrum? I think some discussion on this point would be valuable. The statement: the present findings do not confirm the reports suggesting that cognitive problems are less common in patients with RRMS’ warrants clarification: less common than what? I would recommend greater care in talking about ‘depressive symptoms’ (which is what is measured in the study) rather than depression.

Author Response

Dear Reviewer,

Please find an attached file with our answers to your comments.

Thank you,

Sincerely,

authors

Reviewer 2 Report

Here, the authors describe the results of a matched case-control study of 22 RRMS in remission and 22 controls, assessing the frequency and distribution of executive cognitive dysfunction and how these relate to depression and disease duration. The authors found excesses of dysfunction among cases compared to controls across all modes of assessing executive function. Aside from three measures of executive function (MoCA, category test, and phonemic fluency), no relationships with disease duration or depression were seen. From these results the authors suggest testing of executive function should be a component of routine assessment in RRMS.

I have some various comments which I hope the authors will consider, as below:

  • The Introduction is quite long and meandering. Some shortening is suggested.
  • Info about the cases and controls’ demographics should be in the Results, not the Methods.
  • It is stated in the text that all cognitive tests were worse in the cases than controls. However, the Semantic fluency test would seem to be better for the cases? Please clarify.
  • I am unclear how the matching was applied in this analysis as simple Chi-square and correlation tests were undertaken.
  • I am unclear how deficits was defined in Table 2. Higher number of errors or longer time to complete would seem definable as a deficit but number of words? How was deficit defined – was there some cutoff or was this just defined as less than their control or? Please specify in Methods and Table 2.
  • That disease duration was negatively correlated with MoCA and positively with Category test is counterintuitive as it would suggest longer disease duration is associated with better cognitive function. Please speak to this.
  • I would suggest that these results are not sufficient to indicate any alternations to care in people with MS or the adoption of psychological screening amongst RRMS. If replicated in other and larger samples, maybe, but at best the authors can say “subject to replication” such screening might be implemented.

Other comments:

  • Define abbreviations at first use and then use that abbreviation consistently thereafter, e.g., CNS, RRMS, EF.
  • At line 75, what does “Other problems” entail? Please specify.
  • At line 96, suggest recommendations for future research should be in Discussion, not Introduction.
  • At line 100, suggest clearly specifying that executive function in RRMS was assessed as compared to matched healthy controls.
  • The research questions at lines 104-108 could be condensed.
  • There is no point to presenting statistics for all RRMS and controls combined. Just present characteristics for each. Also, the 44 participants were not divided into two groups; cases and controls were recruited separately.
  • Regarding the RRMS being in remission, was this an inclusion criterion or did they all just happen to be in remission? I presume the former but specify.
  • Regarding the exclusion criteria of participants not having head injuries, physical health problems, or addictions, how were these assessed and what defined an exclusionary level? Also, how many such cases and controls were excluded?
  • Regarding terminology, the category fluency test is alternatively referred to as Semantic fluency and Category test. Please pick one and be consistent.
  • The MoCA has a cutoff indicative of poor function. Do the other tests have cutoffs indicative of poor function and if so, specify.
  • Regarding education amongst the controls, 11 completed secondary but what of the other 11?
  • In Table 1, please also provide the range.
  • In Table 1, suggest BDI-II should be located either at start of end of the table rather than amidst the cognitive tests.
  • Suggest terminology should be frequency, not prevalence.
  • Don’t use the term, confirm. Epidemiological studies don’t confirm, they just suggest, support, or indicate.

Author Response

Dear Reviewer,

Please find an attached file with our responses to your comments.

Thank you,

Sincerely,

authors

Round 2

Reviewer 2 Report

Thanks for your responses. Happy for this to proceed.